# The Bioanalytical and Biomedical Applications of Polymer Modified Substrates

**DOI:** 10.3390/polym14040826

**Published:** 2022-02-21

**Authors:** Guifeng Liu, Xudong Sun, Xiaodong Li, Zhenxin Wang

**Affiliations:** 1Department of Radiology, China-Japan Union Hospital of Jilin University, Xiantai Street, Changchun 130033, China; gfliu@jlu.edu.cn (G.L.); xiaodong20@mails.jlu.edu.cn (X.L.); 2State Key Laboratory of Electroanalytical Chemistry, Changchun Institute of Applied Chemistry, Chinese Academy of Sciences, Changchun 130022, China; sunxudong@ciac.ac.cn; 3School of Applied Chemistry and Engineering, University of Science and Technology of China, Jinzhai Road, Hefei 230026, China

**Keywords:** branched polymers, polymer brushes, polymer hydrogels, substrates, bioanalytical and biomedical applications

## Abstract

Polymers with different structures and morphology have been extensively used to construct functionalized surfaces for a wide range of applications because the physicochemical properties of polymers can be finely adjusted by their molecular weights, polydispersity and configurations, as well as the chemical structures and natures of monomers. In particular, the specific functions of polymers can be easily achieved at post-synthesis by the attachment of different kinds of active molecules such as recognition ligand, peptides, aptamers and antibodies. In this review, the recent advances in the bioanalytical and biomedical applications of polymer modified substrates were summarized with subsections on functionalization using branched polymers, polymer brushes and polymer hydrogels. The review focuses on their applications as biosensors with excellent analytical performance and/or as nonfouling surfaces with efficient antibacterial activity. Finally, we discuss the perspectives and future directions of polymer modified substrates in the development of biodevices for the diagnosis, treatment and prevention of diseases.

## 1. Introduction

Polymers, including synthetic polymers, semi-synthetic polymers and biopolymers, have been demonstrated as a valuable tool for tuning the physicochemical properties of a surface [1,2,3,4,5,6,7,8,9,10,11,12,13,14,15,16,17]. Polymer modified substrates have been extensively applied in various fields such as biomedicine, food safety, environment protection and microelectronics, because the polymers with high molecular weight provide multiple options for the immobilization of bioactive molecules with diverse functionalities and various labeling probes suitable for different detection principles [3,4,5,6,7,8,9,10,11,12,13,14,15,16,17]. In the presence of properly selected reactive groups, including hydroxyl, carboxyl, aldehyde and amine on polymer chains, bioactive compounds can be easily immobilized on the polymer surfaces through covalent reactions and non-covalent interactions. For instance, antibodies can be efficiently immobilized on the surface of hydrophilic polymer brush, poly(2-hydroxyethyl methacrylate-co-2-carboxyethyl acrylate) (p(HEMA-co-CEA)) modified substrate through a simple 1-ethyl-3-(3-dimethylaminopropyl)-carbodiimide and N-hydroxysuccinimide (EDC/NHS) chemistry [18]. Furthermore, the native bioactivities of biological compounds can be preserved by reducing steric constraints and shielding the compounds from hydrophobic surface-induced denaturation when these compounds are tethered to a solid substrate via a polymer [4,15]. In addition, the polymer brushes can not only increase the loading capacity of antibodies on the substrate surface, but can also provide an antifouling layer on the hydrophobic substrate [14,15,16]. These phenomena result in the significant improvement of the analytical performance of biosensor.

Careful selection of a constructing technique is a prerequisite for the successful modification of a substrate with a particular polymer. Several surface methods/strategies, including wet chemical, self-assembly, ionized gas treatments and UV irradiation, have been developed to construct polymer modified substrates with reactive sites [4,19,20,21]. These methods/strategies have their own inherent advantages and disadvantages. For example, the wet chemical approach can be achieved in most laboratories without specialized equipment, which allows for surface functionalization of two- and three-dimensional (2D- and 3D-) substrates. However, the wet chemical method exhibits poor specificity, which may not produce repeatable polymer layer modified substrate on a large scale. The self-assembly strategy can generate an ordered, single molecular polymer layer on the surfaces of inorganic substrates such as glass or silicon by using organosilanes as crosslinkers, and on copper, silver or gold by the formation of metal-sulfur covalent bonds. Unfortunately, the self-assembled polymer monolayer can be decomposed under harsh experimental environments, including high temperatures and extremely high or low pH values. Ionized gas treatments, such as oxygen plasma, corona discharge and flame treatment, can generate finely controlled polymer layer on substrate, but they normally require expensive experimental facilities and well-trained technicians [4,19,20,21]. The UV irradiations can be used to generate reactive sites on polymer surfaces and initiate UV-induced graft polymerization on the substrates [21]. The amount and depth of reactive sites can be adjusted by varying the parameters of UV irradiation such as wavelength and intensity of light and irradiation time. However, the treatment consistency of UV irradiation is strongly affected by the optical properties of polymer.

For bioanalytical and biomedical applications, the bioactive compounds were normally immobilized on the polymer modified substrates through covalent reactions, ligand–receptor pairing (e.g., biotin–avidin) and non-covalent adsorptions (e.g., electrostatic interactions and hydrogen bonds) [4,11,15]. Non-covalent adsorption is a simple process with the benefits of time saving and low cost. However, the non-covalent binding of a bioactive compound with polymer is not strong enough to yield stable functionalized surfaces capable of withstanding the harsh experimental conditions of subsequent biological reactions, resulting in strong non-specific reactions and poor reproducibility. The biotin–avidin (streptavidin) interaction is attractive in surface bioconjugations because of the strong affinity of biotin and avidin (streptavidin) and the number of biotinylated reagents available. Although the covalent attachment is more complex than the electrostatic interaction or physical adsorption immobilization techniques, the covalent binding of bioactive compounds with polymer offers high stability and is demonstrated to be quite robust in complex reaction conditions. In particular, the covalent immobilization can be used to increase the bioactivity of a biomolecule and prevent its metabolism.

Polymer modified substrates have been utilized with the goal of developing specific applications of immobilized biomolecules within a wide range of scientific disciplines. This review focuses on their medical applications as biosensors with excellent analytical performance and/or as nonfouling surfaces with efficient antibacterial activity. The purpose of this review is to try to introduce readers to a general view on the recent development of branched polymer-, polymer brush- and polymer hydrogel-modified substrates and their representative applications.

## 2. Branched Polymers

Various dendrimers, including polyamidoamine (PAMAM), poly(propylene imine) (PPI), poly(ester amides) and phosphorus dendrimers, have been synthesized since the hyperbranched polymers were first defined by Flory in 1952 [22,23,24,25]. Some of these highly branched polymer modified substrates have been extensively used to construct biosensors and/or as drug carriers because polymers with different core structures, terminal groups, generations and grades of purity are commercially available [1,11,12,25,26,27,28,29,30,31,32,33,34,35,36,37,38,39,40]. For instance, Idris et al. developed an electrochemical (EC) immunosensor for the detection of alpha-feto protein (AFP) through the immobilization of anti-AFP antibodies on the gold nanoparticles (AuNPs) and the generation of a 3 PPI dendrimer co-modified glassy carbon electrode (GCE) [31]. The as-prepared immunosensor exhibited a wide concentration range, from 0.005 to 500 ng mL^−1^, with low detection limits (LODs) of 0.0022 ng mL^−1^ (square wave voltammetry (SWV) measurement) and 0.00185 ng mL^−1^ (electrochemical impedance spectroscopy (EIS) measurement). In addition, the EC immunosensor exhibited good stability over a period of 2 weeks, when it was stored at 4 °C. Gu et al. developed a regenerated EC biosensor combined with an in vivo microdialysis system by using hyperbranched polyethyleneimine (hPEI) as a regenerated recognition unit for Cu^2+^ (as shown in Figure 1) [35]. The EC biosensor was capable of determining Cu^2+^ with a linear range from 0.05 to 12 μmol L^−1^ and a low LOD of 13 nmol L^−1^. The EC biosensor can be easily regenerated by thr dissociation of Cu^2+^ and Cu^+^ ions using ethylenediaminetetraacetic acid (EDTA) disodium salt. The EC biosensor has been successfully employed for the repetitive analysis of Cu^2+^ in rat brains under global cerebral ischemia/reperfusion events. Hao et al. developed a microfluidic system for the detection of E. coli O157:H7 through the immobilization of aptamers against E. coli O157:H7 on the surface of a generation 7 PAMAM modified polydimethylsiloxane (PDMS) microfluidic channel [39]. Due to a significant increase in the amount of aptamers available on the microfluidic channel surface, the as-developed microfluidic system exhibited a low LOD of 10^2^ cells mL^−1^. Very recently, Tsekeli et al. developed an aptasensor based on a PPI dendrimer-carbon nanofiber nanocomposite (CNFs-PPI) immobilized GCE for the detection of bisphenol A (BPA) [41]. The amino modified aptamers were immobilized on the CNFs-PPI platform by covalent bonding using glutaraldehyde as a cross-linker. The as-proposed aptasensor (GCE/CNFs-PPI/NH-Apt) was able to selectively detect BPA in the range of 1 nmol L^−1^ to 10 nmol L^−1^, with LODs of 0.03 nmol L^−1^ and 0.06 nmol L^−1^ obtained from differential pulse voltammetry (DPV) and EIS, respectively. The practicability of the GCE/CNFs-PPI/NH-Apt was demonstrated by the detection of BPA in spiked water samples, which were stored in a plastic (polycarbonate) bucket. Recoveries between 91.8% and 100.3% were obtained.

Because of its unique redox activity, Ferrocene (Fc) has been successfully used to improve the performance of EC biosensors through the elimination of most electro-active interferences [42,43]. However, it is difficult to immobilize/adsorb Fc onto the surface of an electrode [44]. Branching conjugation of Fc with polymer enables the circumvention of this drawback [44,45,46,47,48,49]. For instance, Li et al. synthesized a water soluble Fc-terminated hyperbranched polyurethane (HPU-Fc), which was used for the fabrication of a non-enzymatic glucose sensor through an electrodeposition process [47]. The as-developed EC sensor showed a good response to glucose concentration with good stability, favorable accuracy and high selectivity. Kowalczyk et al. developed an EC immunosensor for the selective detection of C-reactive proteins (CRP) in blood samples by using branched polyethylenimine (PEI) functionalized with Fc residues (PEI-Fc) as a recognition layer, which allows: (i) covalent binding of an antibody in its most favorable orientation and (ii) voltammetric detection of the CRP [48]. The PEI-Fc formed a thin, stable and reproducible layer on the electrode surface through the electrodeposition process. The as-proposed EC immunosensor exhibited a linear range from 1 to 5104 ng mL^−1^, with low LODs of 0.5 (DPV measurement) and 2.5 ng mL^−1^ (ESI measurement), which has been successfully employed for the detection of CRP in rat blood samples. Gan et al. developed an EC biosensor for prolonged continuous monitoring of free flap failure caused by vascular occlusion after reconstructive surgery [49]. In this case, FFc-containing chitosan-cografted-branched PEI redox conjugates (CHIT-Fc-co-BPEI-Fc) were used as a pH-tuneable matrix for the attachment of glucose oxidase and lactate oxidase on an electrode surface, respectively. The as-developed glucose oxidase-sensor and lactate oxidase -sensor exhibited good sensitivities, and were found to be 2.89 (± 0.06) μA/(mmol L^−1^) (glucose oxidase) and 2.95 (± 0.19) μA/(mmol L^−1^) (lactate oxidase), with LODs of 0.047 mmol L^−1^ (glucose oxidase) and 0.172 mmol L^−1^ (lactate oxidase), respectively.

Branched biopolymers, including branched peptides (also known as Y-shape pep-tides), branched DNAs and branched oligosaccharides modified substrates, have also been used to fabricate biosensors with high analytical performance [50,51,52,53,54,55,56,57]. Luo’s group has been developing a series of branched zwitterionic peptide-based EC biosensors for the detection of various targets such as proteins, COVID-19 nucleic acid and cells in the complex biological samples [51,52,53,54]. For example, they fabricated an antifouling interface through the covalent conjugation of branched zwitterionic peptides onto an electrodeposited polyaniline film [51]. The branched peptide modified surface exhibited low adsorption of nonspecific proteins and cells. After the immobilization of the mucin1 protein (MUC1) aptamer as the recognition element, the as-developed EC aptasensor exhibited a linear range of 50 to 10^6^ cells mL^−1^, with low LOD of 20 cells mL^−1^, which was used successfully to selectively detect MUC1-positive MCF-7 breast cancer cells in complex samples. Recently, they developed an antifouling EC biosensor based on all-in-one branched peptides that combine anchoring, doping, antifouling and recognizing functions, which were immobilized onto gold nanoparticle modified GCEs (as shown in Figure 2) [52]. After the electrodeposition of a conducting polymer (poly(3,4-ethylenedioxythiophene), PEDOT) on the doping region of a branched peptide for enhancing the interface conductivity, the as-developed EC biosensor exhibited five orders of magnitude dynamic range (from 0.1 ng mL^−1^ to 10 mg mL^−1^ IgG), high stability, excellent selectivity, very low LOD (45 pg mL^−1^ IgG) and acceptable accuracy for serum sample analysis. Jeong et al. fabricated a hybrid film for enhancing the human pluripotent stem cells (hPSCs)-specific EC signals using immobilized AuNPs and branched arginyl-glycyl-aspartic acid (RGD) peptides onto the gold electrode [55]. By taking advantage of AuNPs and branched arginyl-glycyl-aspartic acid (RGD) peptides to increase the adhesion as well as conductibility of hESCs, the as-developed EC biosensor exhibited a linear range of 25,000 cells to 890,000 cells. Given the advantages of its being enzyme-free and high-order growth kinetic, its high sensitivity and its simple operation, the nonlinear hybridization chain reaction (HCR) is regarded as a powerful signal amplifier for the detection of various biomarkers [58,59]. Recently, Jia et al. developed an ultrasensitive EC biosensor for the detection of specific DNA based on nonlinear HCR by triggering the chain-branching growth of DNA dendrimers on the surface gold electrode [58]. Because of the high-order growth kinetic of DNA dendrimers on an electrode surface by HCR, the as-developed EC biosensor exhibited low LOD of 0.4 fmol L^−1^ and was capable of discriminating single nucleotide polymorphism (SNP) among concomitant DNA sequences. Some examples of the bioanalytical and biomedical applications of branched polymers are summarized in Table 1.

## 3. Polymer Brushes

Polymer brushes are surface-tethered polymer chains forming an extremely thin polymer film, which are normally synthesized through surface-initiated atom-transfer radical polymerization (SI-ATRP) [2,14,60,61,62]. Polymer brushes can significantly alter surface properties because it is easy to introduce massive functional groups to them [61]. Currently, polymer brushes have been extensively used for anti-biofouling in biosensors and biomedical equipment, and hold great promise for the development of the next generation of biosensors and diagnostic devices [2,14,60,61,62,63,64,65,66,67,68,69,70,71,72,73,74,75,76,77,78,79,80,81,82,83,84,85,86,87,88,89,90,91,92,93,94,95]. For example, hydrophilic macromolecules such as polyethylene glycol (PEG), polyacrylic acid (PAA) and poly(2-hydroxyethyl) methacrylate (PHEMA) have been grafted onto the substrate surface, providing strong resistance to protein and algae adhesion [2,14,60,61,62,63,64,65,66]. In addition, some examples of the bioanalytical and biomedical applications of polymer brushes are summarized in Table 2.

### 3.1. Polymer Brush-Based Biosensors

The polymer brush modified substrates have been employed to construct different biosensors with a high analytical performance [67,68,69,70,71,72,73,74,75,76]. Costantini et al. developed an aptasensor for the detection of Ochratoxin A (OTA) in a beer sample by the immobilization of aptamer on the PHEMA brush modified glass substrate [68]. The stability and specificity of the aptasensor was significantly enhanced by the PHEMA brush. The aptasensor exhibited high sensitivity (0.32 pA L mg^−1^) and low LOD (0.82 mg L^−1^). Greene et al. demonstrated that an anti-adhesive lubricin brush could be utilized to filter and separate small analyte molecules from large, potentially fouling molecules [69]. Therefore, the lubricin brush coated electrode can be used for highly sensitive amperometric/voltametric detection of small electroactive compounds in highly fouling samples (e.g., 50% diluted blood plasma) with minimal, immediate impact upon the electrochemistry. Ferhan et al. fabricated 2D- and 3D- assemblies of gold nanorods (AuNRs) on polymer brush (termed as surface-floating super-aggregates) with high-density and uniform distribution through the immersion of a poly(oligo ethylene glycol methacrylate) brush-modified substrate in an AuNR solution without any form of functionalization [70]. The surface-floating super-aggregates exhibited strong improvement in surface-enhanced Raman spectroscopy (SERS) sensing performance. The surface-floating super-aggregate-based SERS can be used to detect rhodamine 6G at as low as sub-femtomolar levels. Very recently, Yang et al. developed an integrated three-electrode system (ITES) modified with a “liquid-like” PDMS brush for continuously and stably monitoring reactive oxygen species (ROS) in complex fluids [76]. Benefitting from the antifouling of PDMS brush, the sensing performance of as-developed ITES with PDMS (termed as PMITES) could remain stable and free of bacterial attack even after 3 days of incubation with bacteria. In particular, the PMITES enables the continuous recording of ROS levels in bacterial rich fluids with excellent stability over 24 h, which opens new pathways for the continuous and real-time monitoring of biomarkers in complex biofluids.

### 3.2. Polymer Brush-Based Microarrays

Because of their inherent antibiofouling nature, the fabrication of microarrays on the polymer brush modified substrates allows biomolecular immobilization and recognition with low nonspecific interactions, resulting in a significant improvement of the specificity and reproducibility of microarray [77,78,79,80,81,82,83,84,85,86,87,88,89,90,91,92,93,94,95]. In addition, the 3D structure of polymer brush modified substrates provide a high biomolecule immobilization capacity and accessible scaffolds with sufficient space for biomolecule binding, leading to an increase in the sensitivity of microarrays. Sun et al. developed ultra-low fouling microarrays for protein detection, which included functionalizable polycarboxybetaine methacrylate (pCBMA) grafted arrays for immobilization of biomolecular probes and a nonfunctionalizable polysulfobetaine methacrylate (pSBMA) grafted background (as shown in Figure 3) [83]. Both pCBMA and pSBMA highly resist nonspecific protein adsorption. Due to its strong antifouling property, the microarray can detect as low as 10 ng mL^−1^ bovine serum albumin (BSA) in the sample matrix of bovine serum. In addition, the pCBMA and pSBMA modified substrate can be used to fabricate surface-tension droplet arrays for surface-directing cell adhesion and growth. The pCBMA and pSBMA modification provides an excellent antifouling interface in protein and cell microarrays for possible applications in various bioassays and bioengineering. Hou et al. developed a 3D smart binary polymer-brush pattern on the polymer substrate for generating multiple cell microarrays by using the thermo-responsiveness of poly-(N-isopropylacrylamide) (PNIPAM) and the reaction of concanavalin A (Con A) with poly(D-gluconamidoethyl methacrylate) (PGAMA) [85]. The smart binary polymer-brush pattern-based multiple cell microarrays exhibited high versatility and specificity. Very recently, Valles et al. fabricated a novel glycan microarray on the [ethylene glycol dimethacrylate] (EGDMA) and pentaerythritol tetrakis(3-mercaptopropionate) (PETT) copolymer brush, which binds the mannose-specific glycan binding protein, concanavalin A (ConA), with sub-femtomolar avidity [90]. This finding opens a new era in glycobiology, where the detection of glycan-binding proteins (e.g., lectins) meets the requirements of medical and biological events.

The analytical performance of microarray can be further improved by integrating the advantages of nanomaterials and polymer brushes [91,92,93,94,95,96]. Liu et al. developed a sphere-polymer brush hierarchical nanostructure-modified glass slide (termed as PGMA@3D(160) substrate) for fabricating high-performance microarrays through growing a poly(glycidyl methacrylate) (PGMA) brush layer on the 160 nm silica particle-self-assembled monolayer coated glass slide [92]. The as-developed PGMA@3D(160) substrate can provide 3D polymer brushes containing abundant epoxy groups for directly immobilizing various biomolecules such as glycans, DNA and protein detection. As a typical example, the interactions of three monosaccharides (4-aminophenyl β-D-galactopyranoside, 4-aminophenyl β-D-glucopyranoside, and 4- aminophenyl α-D-mannopyranoside) with two lectins (biotinylated ricinus communis agglutinin 120 and biotinylated concanavalin A from Canavalia ensiformis) have been assessed by PGMA@3D(160) substrate-based carbohydrate microarrays. The carbohydrate microarrays exhibited good selectivity, strong multivalent interaction and low LOD in the picomolar range without any signal amplification, making it a promising platform for bioanalytical and biomedical applications. Cruz et al. developed a microarray-based point-of-care (POC) diagnostic device by inkjet-printed antibodies on a polymer brush (poly(oligo(ethylene glycol) methyl ether methacrylate) (POEGMA)) modified gold film (as shown in Figure 4) [93]. By the integration of a sandwich immunoassay microarray within a plasmonic nanogap cavity between gold film and silver nanocubes, the as-developed microarray can detect as low as 0.02 ng mL^−1^ B-type natriuretic peptide (BNP), which is an important biomarker for the prognosis and long-term monitoring of cardiac disease. Jian et al. developed a peptide microarray-based fluorescence assay for profiling multiple matrix metalloproteinases (MMP-1, -2, -3, -7, -9 and -13) activities in the progression of osteosarcoma (OS, a primary malignant bone tumor) by immobilization of different peptide substrates on the poly(glycidyl methacrylate-co-2-hydroxyethyl methacrylate) brush coated zinc oxide nanorod (ZnONR@P(GMA-HEMA) decorated glass slides [95]. The microarray-based fluorescence assay exhibited excellent selectivity and sensitivity, which enables the detection of the activities of cellular secreted MMPs at picomolar level. The result of the peptide microarray-based fluorescence assay demonstrated that the activity pattern of MMPs in serum is positively relevant to the progression of OS.

### 3.3. Infection Resistance of Polymer Brush Modified Substrates

Bacterial adhesion and biofilm formation have a great impact on the service life of medical devices. It is demonstrated that polymer brush coating can efficiently prevent the adhesion of bacteria on the surface [96,97,98,99,100,101,102,103,104,105,106,107,108]. For instance, Ibanescu et al. found that both poly(2-hydroxy ethyl methacrylate) brush and poly(poly(ethylene glycol)methacrylate) brush exhibited strong anti-adhesion capability of *Staphylococcus epidermidis* [96]. Sae-ung et al. demonstrated that the adhesion of *Escherichia coli* on a silicon surface was efficiently prevented by a coating of copolymer of methacryloyloxyethyl phosphorylcholine (MPC) and a methacrylate-substituted dihydrolipoic acid (DHLA) [98]. Su et al. developed an antibiofouling surface through grafted PAA-g-PEG (MW 2000, 6000, and 11,000 Da) on the plastic and elastomer surface [102]. Comparison with the initial substrate, the PAA-g-PEG modified substrate, shows excellent and long-lasting antibiofouling properties to resist the adhesion of algae, demonstrating that the hierarchical comb hydrophilic polymer brushes exhibit strong capacity against the adhesion of marine microorganisms. Wang et al. constructed two types of polymer brushes with different hierarchical structures (termed as polyDVBAPS/poly(HEAA-g-TCS) and poly(DVBAPS-b-HEAA-g-TCS)) through integration of salt-responsive polyDVBAPS (poly(3-(dimethyl(4-vinylbenzyl) ammonio)propyl sulfonate)), antifouling polyHEAA (poly(N-hydroxyethyl acrylamide)) and bactericidal TCS (triclosan) onto single silicon wafer surface [103]. Due to a synergistic effect of the three compatible components, both the polyDVBAPS/poly(HEAA-g-TCS) brush and the poly(DVBAPS-b-HEAA-g-TCS) brush modified surface exhibited excellent antibacterial activity, offering a promising strategy to fabricate next-generation infection-resistant surfaces for various antibacterial applications. Dhingra et al. systematically studied the infection resistance of three polymer brushes, namely PHEMA, poly (poly (ethylene glycol) methacrylate) (PPEGMA) and poly[(2-methacryloyloxyethyl] trimethyl ammonium chloride) (PMETA) on hydroxyl functionalized polyester substrate [104]. Among the three polymer brushes, PMETA exhibited the highest antibacterial activity, with only ~3% and ~4% adherence of *Escherichia coli* and *Staphylococcus aureus*, respectively. Very recently, Wu et al. fabricated a mixed-charge copolymer brush (#1-A) modified polyurethane (PU) catheter by using two oppositely charged monomers, the anionic SPM (3-Sulfopropyl methacrylate) and the cationic AMPTMA ((3-Acrylamidopropyl) trimethylammonium chloride) [106]. The #1-A exhibited 99% reductions against all six Gram-positive and Gram-negative bacteria including methicillin-resistant *Staphylococcus aureus* (MRSA BAA38), methicillin-resistant *Staphylococcus epidermidis* (MRSE 35984), vancomycin-resistant *Enterococcus faecalis* (VRE V583), *Pseudomonas aeruginosa* PAO1, uropathogenic *Escherichia coli* (UTI89) and carbapenem-resistant *Acinetobacter baumannii* (AB-1).

## 4. Polymer Hydrogels

### 4.1. Polymer Hydrogel-Based EC Biosensor

Conducting polymer hydrogels (CPHs) have been extensively used for the development of EC biosensors because they have a large specific surface area, good biocompatibility and 3D continuous conducting network [109,110,111,112,113,114,115,116,117]. For instance, Geleta et al. developed a cost effective, environmentally friendly and disposable EC aptasensor (termed SPCE/PAM/PA/PDA/Apt) for the detection of Aflatoxin B2 (AFB2) through the immobilization of an AFB2 aptamer (Apt) on a conducting porous polyacrylamide/phytic acid/polydopamine (PAM/PA/PDA) hydrogel modified screen printed carbon electrode (SPCE) [112]. The as-developed SPCE/PAM/PA/PDA/Apt exhibited a wide dynamic range from 0.1 pg mL^−1^ to 100 ng mL^−1^ and a low LOD of 0.10 pg mL^−1^, which was successfully used to determine AFB2 in spiked corn extracts. Wang et al. developed an EC biosensor for the detection of human epidermal growth factor receptor 2 (HER2), a well-known breast cancer biomarker, through the fabrication of an antifouling sensing interface based on the conducting polymer poly(3,4-ethylenedioxythiophene) (PEDOT) and a biocompatible peptide (Phe-Glu-Lys-Phe functionalized with a fluorine methoxycarbonyl group, Fmoc-FEKF) hydrogel (as shown in Figure 5) [115]. The as-developed biosensor exhibited a wide linear range from 0.1 ng mL^−1^ to 1.0 μg mL^−1^, with a low LOD of detection of 45 pg mL^−1^, which was capable of detecting HER2 in human serum with good accuracy. Ma et al. developed a polypyrrole and vinyl Fc/mono-aldehyde β-cyclodextrin (β-CD) complex-based EC immunosensor for the detection of motilin [116]. The as-developed EC immunosensor exhibited a wide linear range of 10 pg mL^−1^ to 100 ng mL^−1^, low LOD of 2.73 pg mL^−1^ and a highly sensitive response, with the slope value as high as 31.342, making great sense in the practical diagnosis. Yang et al. developed an EC biosensor by using a conjugated polypyrrole (PPy) hydrogel with conductive sulfonated multi-walled carbon nanotubes (s-MWCNTs) as crosslinking agents [117]. Due to the integration of the advantages of PPy hydrogel and s-MWCNTs, the as-developed EC biosensor enables sensitive in situ detection of biomolecules released from living cells and real-time monitoring of cell proliferation.

### 4.2. Polymer Hydrogel-Based Optical Biosensor

Because polymer hydrogels have the ability to expand/shrink through the absorption (or desorption) of water by external stimuli (e.g., a pH change and temperature change), their volume change could induce a change of optical properties. Therefore, the polymer hydrogels are good candidates for fabricating optical biosensors [118,119,120,121,122,123]. Noh et al. fabricated a biosensor array using an interpenetrating polymer network consisting of photonic film templated from reactive cholesteric liquid crystal (CLC) and urease-immobilized PAA [121]. The as-fabricated dots of biosensor array changed their color at 7.5 × 10^−3^ mol L^−1^ urea, which could be used as a cost-effective and easy visual detection without any sophisticated instruments. Makhsin et al. developed a metal-clad leaky waveguide (MCLW) biosensor by using acrylate based-hydrogel including PEG diacrylate (PEGDA, Mn 700), PEG methyl ether acrylate (PEGMEA, Mn 480), and acrylate-PEG2000-NHS coated 9.5 nm titanium waveguide layer on a glass substrate [122]. Under optimized experimental conditions, the as-developed MCLW biosensor generated a single-mode waveguide signal with a refractive index (RI) sensitivity of 128.61 ± 0.15° RIU^−1^ and LOD of 2.2 × 10^−6^ RIU with excellent signal-to-noise ratio for glycerol detection.

### 4.3. Polymer Hydrogel-Based Microarray

Polymer hydrogels have been extensively used to construct various microarrays for screening biomaterials and biomarkers, studying the interactions of biomolecules and detecting different targets [124,125,126,127,128,129,130,131,132,133,134,135,136,137,138,139,140,141,142,143,144,145,146,147,148]. Jia et al. developed peptide-functionalized hydrogel microarrays for the discovery of culture substrates through light-assisted copolymerization of poly(ethylene glycol) diacrylates (PEGDA) and methacrylated peptides [129]. They found that PMQKMRGDVFSP exhibited high activity to support adhesion and sarcomere formation of hiPSC-derived cardiomyocytes (hiPSC-CMs). The as-developed offers a novel strategy for screening biological ligands to develop biomaterials for stem cell and tissue engineering applications. Using surface plasmon resonance imaging (SPRi) as a detection technology, Zhou et al. developed a protein microarray on a 3D-dextran hydrogel chip surface for the profiling of the interactions between small molecule drugs (or candidates) and target proteins [136]. The as-developed protein microarray on 3D-dextran hydrogel exhibited good quality and uniformity (CV = 10.3%, n = 48), which could be used as a label-free high-throughput technology for screening/forecasting side effects of drugs (or candidates) and identifying personalized medicine, etc. Scherag et al. developed an antibody microarray on the polydimethylacrylamide with a 5 mol % 4-methacryloyloxobezophenone (PDMAA-5%BP) modified substrate [139]. The as-fabricated PDMAA-5%BP modified substrate exhibited protein-repellent properties for avoiding unspecific adsorption of analyte molecules during the assay, resulting in a simplification of the assaying procedure, a reduction in background signals and an improvement of the detection sensitivity of the microarray immunoassays. Díaz-Betancor et al. developed a dextran-based nucleic acid microarray by the immobilization of capture probes in dextran polymer hydrogel via a lightinduced thiol–acrylate coupling reaction and polymerization [140]. This approach enables the detection of as low as 2.92 pg μL^−1^ miR-182, which is much lower than the normal amount of circulating miRNA (within the ng μL^−1^ range). Tian et al. developed a lectin microarray based on the polyacrylamide hydrogel for screening the high expression of glycans on the colorectal cancer (CRC) cell surface and to identify new lectin biomarkers for CRC [141]. The immobilized lectins on PAAM hydrogel provide multivalent binding scaffolds to the cellular glycans, resulting in increased binding affinity and the selectivity of lectin with glycan. They demonstrated that Uelx Europaeus Agglutinin I (UEA-I) could be used as new biomarker for CRC subtype SW480. The finding opens up possibilities for discovering lectin biomarkers toward various biomedical applications, including cancer diagnosis and therapy. Recently, Hageneder et al. developed a responsive hydrogel binding matrix for dual signal amplification in fluorescence affinity biosensors and peptide microarrays by using a terpolymer modified metallic sensor surface [148]. The terpolymer, derived from poly(Nisopropylacrylamide) (pNIPAAm), was arranged in arrays of sensing spots and employed for the specific detection of human IgG antibodies against the Epstein−Barr virus (EBV) in diluted human plasma by using a set of peptide ligands. The possibility of using the temperature-induced collapse of the pNIPAAm hydrogel for compacting the captured analytes yielded a route for efficient in situ fluorescence measurements with the combined enhancement factor > 10^3^ under realistic conditions and complex samples. The as-developed offers a new platform for rapid and sensitive fluorescence monitoring of biomolecular interacting events in conjunction with external actuation (e.g., temperature). For the convenience of readers, some examples of the bioanalytical and biomedical applications with polymer hydrogels are summarized in Table 3.

### 4.4. Polymer Hydrogel-Based Bioelectronics

Polymer hydrogels such as PDMS hydrogel and PEG hydrogel have been extensively applied for the construction of stretchable electronic devices because PDMS offers a number of attractive properties, including biocompatibility, ease of process, optical transparency and a moderate elastic modulus [149,150,151,152,153]. Zhang et al. fabricated high-resolution metal microelectrodes with a channel length as short as 5 μm on PDMS by using ultrathin Parylene film (2 μm thick) transfer patterning (as shown in Figure 6) [149]. A fully stretchable organic EC transistor (OECTs) was achieved by combining transfer patterning of metal electrodes with orthogonal patterning of the poly(3,4-ethylenedioxythiophene) doped with polystyrenesulfonate (PEDOT:PSS) on a prestretched PDMS substrate and a biocompatible “cut and paste” hydrogel. Aggas et al. developed a complex hydrogel disk for the attachment and differentiation of PC-12 neural progenitor cells by mixing PEDOT:PSS in poly(2-hydroxyethyl methacrylate-co-polyethyleneglycol methacrylate) p(HEMA-co-EGMA) [152]. The experimental results demonstrated that as-developed hydrogel disk array exhibited good electroconductivity and low cytotoxicity.

## 5. Conclusions and Outlook

Polymer modified substrates show great potential for bioanalytical and biomedical applications. For instance, various EC biosensors have been developed for the accurate de-termination of different analytes with high sensitivity and good selectivity through efficient immobilization of different biorecognition units (i.e., biomolecular probes) on the conductive polymers and/or dendrimers modified electrode surfaces. Polymer-based microarray has been demonstrated as a powerful tool to study biomolecular interactions, the relationship between material physiochemical properties and stem cell responses, etc. Different anti-adhesive hydrophilic polymer brushes have been extensively employed to build antifouling surfaces for the prevention of microbial adhesion and protein adsorption. Polymer hydrogels have been used for the construction of bioelectronics with high biocompatibility. Because of the rich variety of polymer materials, the ability to create a well-defined surface with specific functionality opens new perspectives in bioanalytical and biomedical fields.

The generation of a high quality polymer modified surface is very complex, which requires a deeper understanding of the surface science. It is important to thoroughly consider the selection and design of the polymer for each specific application because the physicochemical properties of polymers play an essential role in surface functionality. The biosensors and microarrays require large numbers and a high reactivity in the molecules in the surface layer, which might be satisfied by increasing the network structure of hydrogel and the density of functional groups in polymer brushes and hyperbranched polymers. The surface antibacterial capability can be improved significantly by the integration of a hydrophilic component and a bactericidal component into one polymer brush through grafting polymerization. Biocompatibility is important factor in the construction of polymer modified substrate for cell adhesion and in vivo applications. Stimulus-responsive polymer may help to develop self-reporting biosensors and a self-cleaning device. In addition, the long-term stability and the failure conditions should be carefully explored when the polymer modified substrate is used to fabricate regeneration devices and wearable bioelectronics. These developments will benefit from a close collaboration among scientists in the fields of material, chemistry, biology and surface science, which will have a strong impact on the diagnostics market.

## Figures and Tables

**Figure 1 polymers-14-00826-f001:**
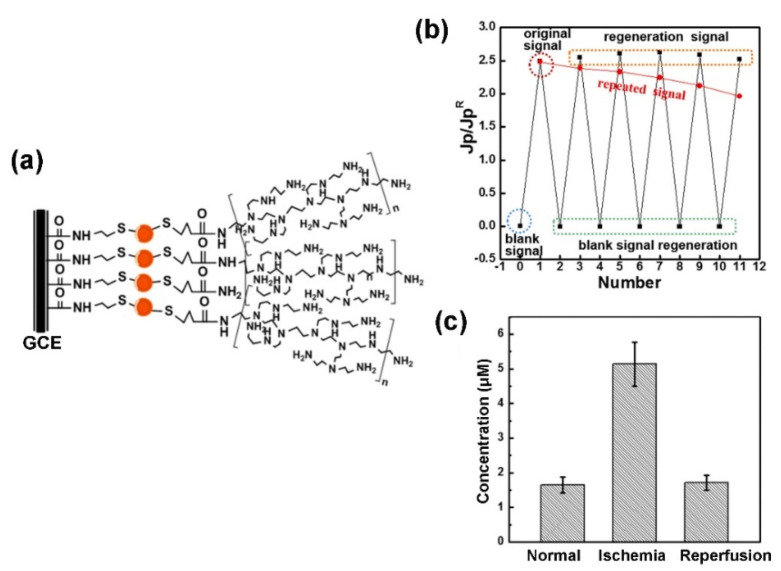
(**a**) Schematic representation of hPEI modified GCE, (**b**) EDTA disodium salt-mediated on-line regeneration of hPEI modified GCE in artificial cerebrospinal fluid (aCSF) solution containing 4 μM Cu^2+^ and (**c**) concentrations of Cu^2+^ in rats under normal, ischemia and reperfusion conditions. Data are expressed as mean±standard error of mean (SEM, n  =  6). (Adapted from Gu et al. 2019 [35], copyright 2019 Elsevier B.V. All rights reserved and reproduced with permission).

**Figure 2 polymers-14-00826-f002:**
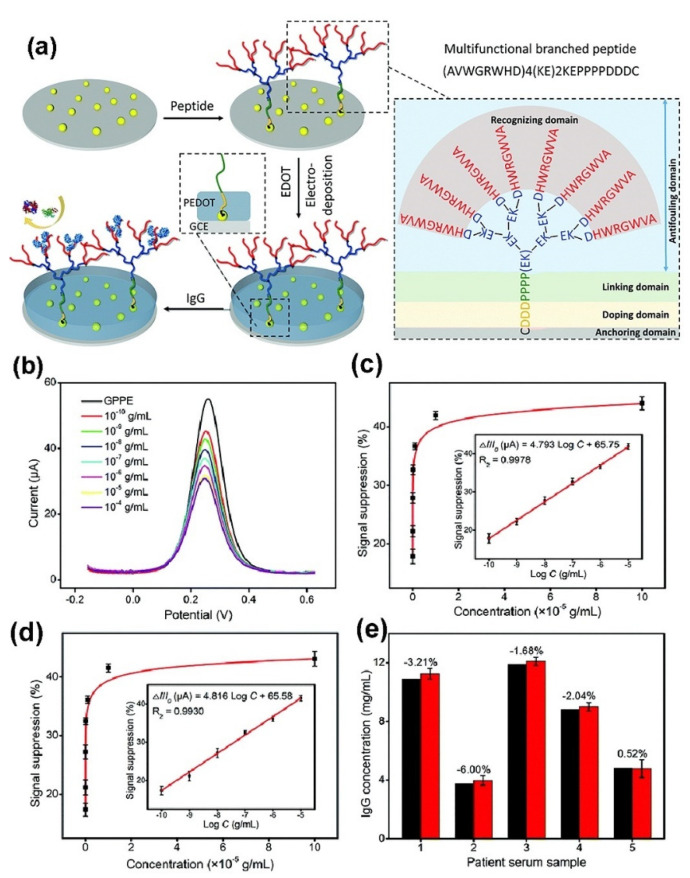
(**a**) Schematic representation of the preparation of an all-in-one branched peptide-based biosensor, (**b**) DPV responses of the biosensor to target IgG, peak current signal suppression of the biosensor after incubation with various concentrations of IgG in the absence (**c**) and presence (**d**) of 1.0 mg mL^−1^ HSA in PBS and (**e**) assay results of clinical serum samples using the developed and reference (black column) methods. The differences (%) in these two methods are labelled. The insets of (**c**) and (**d**) show the corresponding calibration curves. (Adapted from Liu et al. 2021 [52], copyright 2021 The Royal Society of Chemistry. All rights reserved and reproduced with permission).

**Figure 3 polymers-14-00826-f003:**
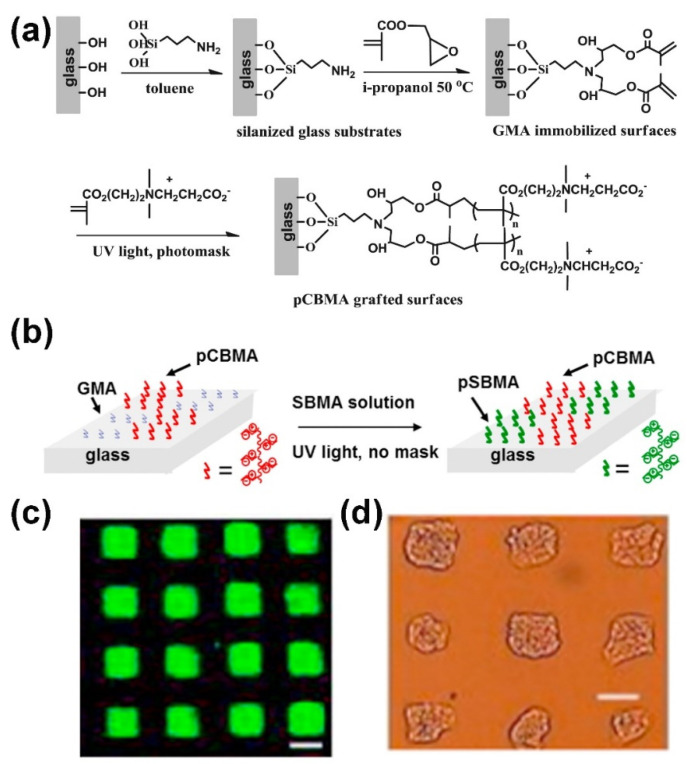
(**a**) Synthetic strategies for the preparation of pCBMA functionalized surfaces on glass substrates, (**b**) schematic diagram of microarrays for cell patterning (left) and protein analysis (right) by photopolymerization of zwitterionic polymers on glass substrates, (**c**) detection of fluorescently labeled BSA on a pCBMA and pSBMA constructed antifouling microarray (scale bar: 100 µm), and (**d**) representative microscopy images of MCF-7 cells patterned and grown on GMA grafted squares for 2 days (scale bar: 100 µm). (Adapted from Sun et al. 2018 [83], copyright 2017 Elsevier B.V. All rights reserved. and reproduced with permission).

**Figure 4 polymers-14-00826-f004:**
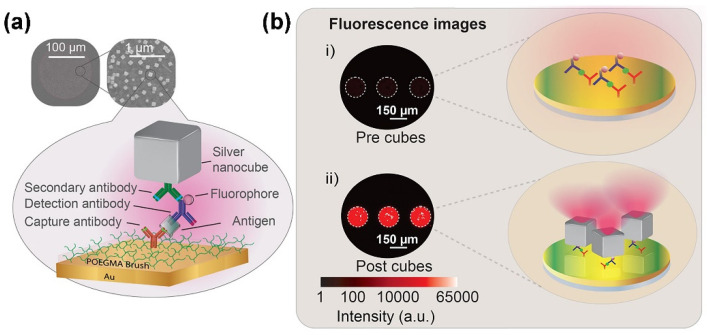
(**a**) Schematic representation of a microarray-based point-of-care (POC) diagnostic device by inkjet-printed antibodies on a polymer brush (POEGMA) modified gold film. Silver nanocubes were adhered to the assay for enhancement of fluorescence intensity by using an interfacial poly(allylamine hydrochloride) (PAH) layer. (**b**) The attachment of silver nanocubes leads to 216-fold fluorescence enhancement of the capture spots for a 1.9 ng/mL BNP concentration and a 151-fold increase when compared to a glass control at the same concentration. (Adapted from Cruz et al. 2020 [93], copyright 2020 American Chemical Society All rights reserved and reproduced with permission. The CC-BY-NC-ND license does not allow third-parties to create derivative works such as translations and only permits other types of use for noncommercial purposes. For more details on the Creative Commons license, please visit the Creative Commons website: www.creativecommons.org (The accessed date of the link: 22 January 2022)).

**Figure 5 polymers-14-00826-f005:**
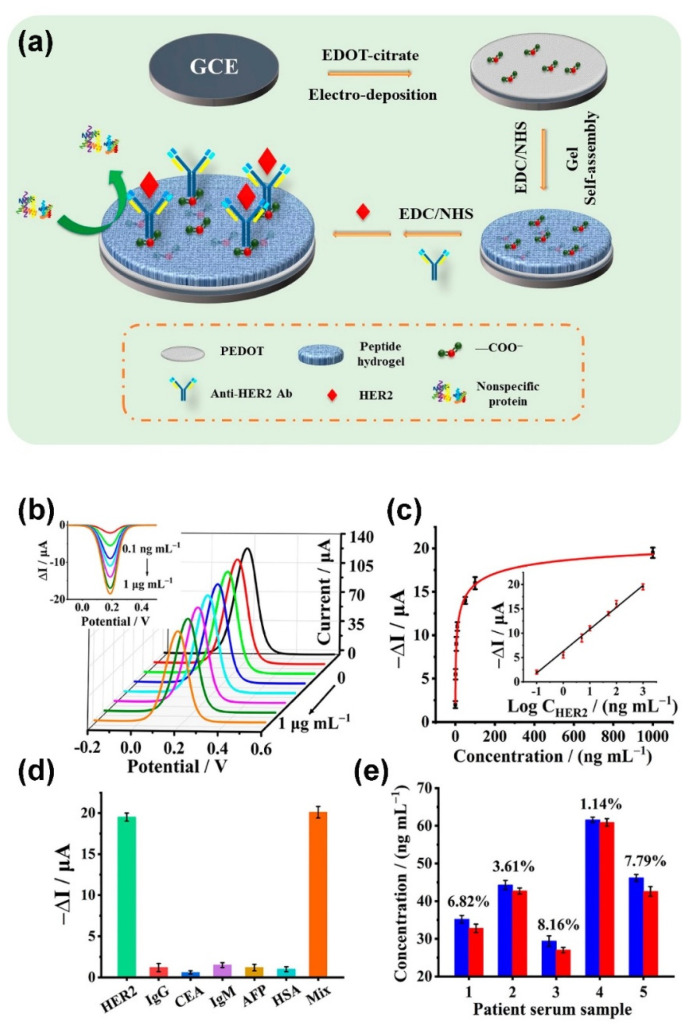
(**a**) Schematic description of the fabrication process of the PEDOT/peptide hydrogel-based HER2 sensor. (**b**) DPV responses of the HER2 biosensor after incubation in HER2 for a range of concentrations. Inset: DPV responses after background removal. (**c**) Corresponding response signal change of the HER2 biosensor. Inset: calibration curve of the HER2 biosensor. (**d**) Responses of the HER2 biosensor to 0.1 mg mL^−1^ of IgG, CEA, IgM, AFP, HSA and a solution mixture (Mix) of those proteins, respectively. (**e**) HER2 determination of breast cancer patient serum utilizing a commercial ELISA Kit (blue) and the developed biosensor (red). Error bars show the standard deviations of three repeated measurements. (Adapted from Wang et al. 2021 [115], copyright 2021 American Chemical Society. All rights reserved and reproduced with permission).

**Figure 6 polymers-14-00826-f006:**
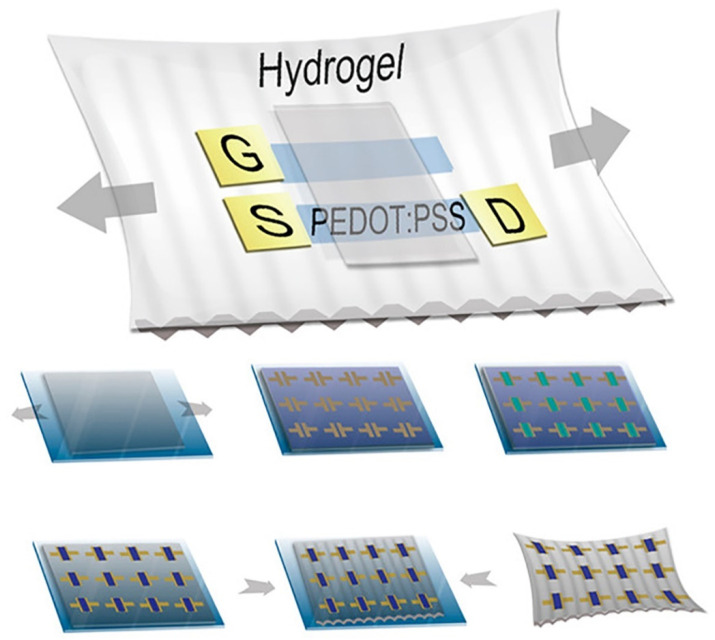
Schematic description of fabrication of stretchable organic EC transistors by using the conducting polymer PEDOT doped with PSS (PEDOT:PSS) on a prestretched PDMS substrate and a biocompatible “cut and paste” hydrogel. (Adapted from Zhang et al. 2017 [149], copyright 2017 American Chemical Society All rights reserved and reproduced with permission).

**Table 1 polymers-14-00826-t001:** Some examples of the bioanalytical and biomedical applications of branched polymers modified substrates.

Polymers	Modification Methods	Detection Method	Analytes	Linear Ranges	Limit of Detection	Ref.
Poly(propylene imine)	Electrodeposition	SWV and EIS	AFP	0.005 to 500 ng mL^−1^	0.0022 ng mL^−1^ (SWV) and 0.00185 ng mL^−1^ (EIS)	[31]
Polyethyleneimine	Covalent modification	DPV	Cu^2+^	0.05 to 12 μM	13 nM	[35]
Polyamidoamine	Covalent modification	Fluorescence	*E. coli* O157:H7	-	1 × 10^2^ cells mL^−1^	[39]
Poly(propylene imine)	Covalent modification	EIS, DPV and CV	BPA	1 to 10 nM	0.03 nM (DPV) and 0.06 nM (EIS)	[41]
Polyurethane	Drop-casting	CV	Glucose	0.1 to 40 mM	60 μM	[47]
Polyethylenimine	Electrodeposition	CV and ESI	CRP	1 to 5 × 10^4^ ng mL^−1^	0.5 ng mL^−1^ (CV) and 2.5 ng mL^−1^ (ESI)	[48]
Polyaniline	Electrodeposition	DPV	MCF-7	50 to 1 × 10^6^ cells mL^−1^	20 cells mL^−1^	[51]
Poly(3,4-ethylenedioxythiophene	Electrodeposition	DPV	IgG	0.1 to 1 × 10^7^ ng mL^−1^	4.5 × 10^−2^ ng mL^−1^	[52]
Branched arginyl-glycyl-aspartic acid peptides	Covalent modification	DPV	Human embryonic stem cells	2.5 × 10^4^ to 8.9 × 10^4^ cells	2.5 × 10^4^ cells	[55]

SWV: square wave voltammetry; ESI: electrochemical impedance spectroscopy; DVP: differential pulse voltammetry; CV: cyclic voltammetry; BPA: bisphenol A; CRP: C-reactive protein.

**Table 2 polymers-14-00826-t002:** Some examples of the bioanalytical and biomedical applications of polymer brushes modified substrates.

Polymers	Modification Methods	Detection Method	Analytes	Linear Ranges	Limit of Detection	Ref.
Poly(2-hydroxyethtyl methacrylate)	SI-ATRP	Electrochemiluminescence	OTA	0 to 10 ng mL^−1^	0.82 ng mL^−1^	[68]
Poly(oligo ethylene glycol methacrylate)	SI-ATRP	SERS	Rhodamine 6G	-	0.1 fM	[70]
Polydimethylsiloxane	Covalent modification	CV	ROS	-	-	[76]
Polycarboxybetaine methacrylate and polysulfobetaine methacrylate	SIPP	Fluorescence	BSA	-	10 ng mL^−1^	[83]
Poly(glycidyl methacrylate)	SI-ATRP	Fluorescence	biomolecules	-	-	[92]
Poly(oligo(ethylene glycol) methyl ether methacrylate	SI-ATRP	Fluorescence	BNP	-	0.02 ng mL-1	[93]
Poly(glycidyl methacrylate-co-2-hydroxyethyl methacrylate)	SI-ATRP	Fluorescence	MMPs	-	10 pM (MMP-1)	[95]

SI-ATRP: Surface-initiated atom-transfer radical polymerization; OTA: ochratoxin A; BSA: bovine serum albumin; ROS: reactive oxygen species; SIPP: surface-initiated photo-polymerization; BNP: B-type natriuretic peptide; MMPs: matrix Metalloproteinases.

**Table 3 polymers-14-00826-t003:** Some examples of the bioanalytical and biomedical applications of polymer hydrogels modified substrates.

Polymers	Modification Methods	Detection Method	Analytes	Linear Ranges	Limit of Detection	Ref.
Polyacrylamide and polydopamine	Drop-casting	DPV	Aflatoxin B2	1 × 10^−4^ to 100 ng mL^−1^	1 × 10^−4^ ng mL^−1^	[112]
Poly(3,4-ethylenedioxythiophene)	Self-assembly	DPV	HER2	0.1 to 1 × 10^3^ ng mL^−1^	4.5 × 10^−2^ ng mL^−1^	[115]
Polypyrrole	Drop-casting	EIS and SWV	Motilin	1 × 10^−2^ to 100 ng mL^−1^	2.73 × 10^−3^ ng mL^−1^	[116]
Polypyrrole	Drop-casting	Chronoamperometry	Biomolecules	-	-	[117]
Polyacrylic acid	UV-curing	Absorption spectra	Urea	-	-	[121]
PEG diacrylate, PEG methyl ether acrylate and acrylate-PEG_2000_-NHS	Covalent modification	Single-mode waveguide	Glycerol	-	2.2 × 10^−6^ RIU	[122]
Dextran T-2000	Spin-coating	SPRi	drugs	-	-	[136]
Dextran methacrylate	Photopolymerization	Fluorescence	miR-182	-	2.92 ng mL^−1^	[140]
Polyacrylamide	SI-ATRP	Fluorescence	Glycans	-	-	[141]
Poly(Nisopropylacrylamide)	Spin-coating	Fluorescence	Human IgG antibodies against the Epstein−Barr virus	-	-	[148]

HER2: human epidermal growth factor receptor 2; SPRi: Surface plasmon resonance imaging.

## Data Availability

Not applicable.

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
