# Peer review of "The Bioanalytical and Biomedical Applications of Polymer Modified Substrates"

_polymers, 2022, doi:10.3390/polym14040826_

Round 1

Reviewer 1 Report

Dear Editor, dear Authors, Guifeng Liu et al. submitted a review paper summarizing the recent advances of polymer modified substrates in bioanalytical and biomedical applications, mainly as biosensors and nonfouling surfaces with efficient antibacterial activity. The authors focused in their manuscript on the functionalization using branched polymers, polymer brushes and polymer hydrogels. The authors have also discussed some perspectives and future directions of polymer modified substrates in the development of biodevices for diagnosis, treatment, and prevention of diseases. The manuscript is of interest and well written, results are supported by experimental evidence. Therefore, for my opinion the manuscript can be accepted for publication in Polymers MDPI Journal. I have very few comments to the authors listed below.

Comments to the authors:

I suggest that the authors include a table summarizing the different examples used in this manuscript and their different applications.

Page 1, line 33 “because the polymers with high molarweights…” I suggest that the authors use here polymers with high molecular weight instead of molarweights.

Page 2, lines 60, 61 “Unfortunately, the self-assemble polymer monolayer…” correct self-assemble by self-assembled.

Page 3, line 99 “… generation and grade of purity are commercial availability” correct are commercial availability by are commercially available.

Page 8, adjust the position of the figure 4.

Page 9, lines 328, 329 “demonstrated that the adhesion of Escherichia coli on silicon surface was efficiently prevent by…” use efficiently prevented by instead of prevent by.

Sincerely Yours,

Author Response

1) I suggest that the authors include a table summarizing the different examples used in this manuscript and their different applications.

Response: Thank you very much for your comments.

We have added three tables for summarizing the different examples used in this manuscript and their different applications, shown as Table 1 to 3 in the revised manuscript.

2) Page 1, line 33 “because the polymers with high molarweights…” I suggest that the authors use here polymers with high molecular weight instead of molarweights.

Thank you very much for your suggestion.

We have changed "molarweights" to "molecular weight" in the revised manuscript.

3) Page 2, lines 60, 61 “Unfortunately, the self-assemble polymer monolayer…” correct self-assemble by self-assembled.

Thank you very much for your suggestion.

We have changed "self-assemble" to "self-assembled" in the revised manuscript.

4) Page 3, line 99 “… generation and grade of purity are commercial availability” correct are commercial availability by are commercially available.

Thank you very much for your suggestion.

We have changed "commercial availability" to "commercially available" in the revised manuscript.

5) Page 8, adjust the position of the figure 4.

Thank you very much for your advice.

According to your advice, we have adjusted the position of Figure 4 in the revised manuscript.

6) Page 9, lines 328, 329 “demonstrated that the adhesion of Escherichia coli on silicon surface was efficiently prevent by…” use efficiently prevented by instead of prevent by.

Thank you very much for your suggestion.

We have changed "prevent" to "prevented" in the revised manuscript.

Reviewer 2 Report

This manuscript by Liu et al. reviewing on the applications of polymer modified substrates is attractive. The authors did a lot of efforts to reorganization. The authors introduce branched polymers, polymer brushes and hydrogels.

1. In each section , the fabrication, mechanism and application of the polymer substrates were described well. However, it would be better to raise points or summary their application in one paragraph or table. The knowledge would be enriched.

2. The authors introduce the polymers very clear on the topic. However, the application need reorganized as I sent. Why are the BPA and CRP detected on branch polymers, OTA and ROS monitored on polymer brush, as well as CLC determined on polymer gel? The author may pay more efforts to explain the unique of each polymer for the sample that the polymer are applied.

Author Response

1) In each section, the fabrication, mechanism and application of the polymer substrates were described well. However, it would be better to raise points or summary their application in one paragraph or table. The knowledge would be enriched.

Response: Thank you very much for your comments and advices.

According to your advice, we added three tables for summarizing the different examples used in this manuscript and their different applications, shown as Table 1 to 3 in the revised manuscript.

2) The authors introduce the polymers very clear on the topic. However, the application need reorganized as I sent. Why are the BPA and CRP detected on branch polymers, OTA and ROS monitored on polymer brush, as well as CLC determined on polymer gel? The author may pay more efforts to explain the unique of each polymer for the sample that the polymer are applied.

Thank you for your comment. Unfortunately, there is no general principle for application of polymer in the analytical science. Most of the experiments and results are random. Normally, the branch polymers and polymer brushes were employed for increasing detection sensitivity, while the hydrophilic polymers were used for antifouling. We have addressed this issue in the manuscript. In the future research, it is possible to produce general principle for application of polymer in the analytical science through the cooperation of researchers in different discipline.